# Plasticity of calcium-permeable AMPA glutamate receptors in Pro-opiomelanocortin neurons

Shigetomo Suyama[1,2], Alexandra Ralevski[1], Zhong-Wu Liu[1], Marcelo O Dietrich[1], Toshihiko Yada[2], Stephanie E Simonds[3], Michael A Cowley[3], Xiao-Bing Gao[1], Sabrina Diano[1,4,5], Tamas L Horvath[1,4,5,6]*

[1]Program in Integrative Cell Signaling and Neurobiology of Metabolism, Section of Comparative Medicine, Yale University School of Medicine, New Haven, United States; [2]Division of Integrative Physiology, Department of Physiology, Jichi Medical University, Tochigi, Japan; [3]Biomedicine Discovery Institute, Department of Physiology, Monash University, Clayton, Australia; [4]Departments of Ob/Gyn and Reproductive Sciences, Yale University School of Medicine, New Haven, United States; [5]Department of Neurobiology, Yale University School of Medicine, New Haven, United States; [6]Department of Anatomy and Histology, University of Veterinary Medicine, Budapest, Hungary

*For correspondence: tamas. horvath@yale.edu

Competing interests: The authors declare that no competing interests exist.

**Abstract** POMC neurons integrate metabolic signals from the periphery. Here, we show in mice that food deprivation induces a linear current-voltage relationship of AMPAR-mediated excitatory postsynaptic currents (EPSCs) in POMC neurons. Inhibition of EPSCs by IEM-1460, an antagonist of calcium-permeable (Cp) AMPARs, diminished EPSC amplitude in the fed but not in the fasted state, suggesting entry of GluR2 subunits into the AMPA receptor complex during food deprivation. Accordingly, removal of extracellular calcium from ACSF decreased the amplitude of mEPSCs in the fed but not the fasted state. Ten days of high-fat diet exposure, which was accompanied by elevated leptin levels and increased POMC neuronal activity, resulted in increased expression of Cp-AMPARs on POMC neurons. Altogether, our results show that entry of calcium via Cp-AMPARs is inherent to activation of POMC neurons, which may underlie a vulnerability of these neurons to calcium overload while activated in a sustained manner during over-nutrition.

## Introduction

Proopiomelanocortin (POMC)-producing neurons in the arcuate nucleus (ARC) of the hypothalamus are well known to control satiety and glucose metabolism (*Horvath, 2006*). The ablation of POMC neurons or the depletion of the POMC gene induces marked obesity in animals (*Gropp et al., 2005*; *Smart et al., 2006*; *Xu et al., 2005*; *Yaswen et al., 1999*), while the optical activation of POMC neurons through channelrhodopsin-2 (ChR2) selectively expressed in these neurons diminishes feeding behavior (*Aponte et al., 2011*). POMC neurons integrate peripheral metabolic signals such as leptin, insulin and glucose, and alter other circuits to adjust behavior and autonomic regulation, in part, through melanocortin receptor signaling (*Cone, 2005*; *Gao and Horvath, 2007*; *Horvath, 2006*; *Koch et al., 2015*; *Mountjoy et al., 1992*; *Ollmann et al., 1997*). In addition to peripheral metabolic signals, synaptic transmission is also required to regulate POMC neuronal activity and body weight (*Branco et al., 2016*). Furthermore, recent studies showed that sensory detection of food inactivated Agouti-related protein (AgRP) neurons and activated POMC neurons in the ARC by fast neurotransmission (*Betley et al., 2015*; *Chen et al., 2015*). It was demonstrated that the synaptic input

organization on POMC neurons is plastic following the dynamic changes of the overall energy status of animals (*Horvath, 2006*). In this process, the number and transmission efficacy of glutamatergic synaptic inputs onto POMC neurons are increased by anorexigenic signals such as leptin and estrogen (*Acuna-Goycolea and van den Pol, 2009*; *Gao et al., 2007*; *Horvath and Gao, 2005*; *Horvath et al., 2010*; *Kim et al., 2014*; *Pinto et al., 2004*; *Sotonyi et al., 2010*). Whether or not plastic changes also occur within the synapse in these neurons is not well-defined.

Rapid glutamatergic transmission plays a critical role in the overall function of the central nervous system. The functional regulation of glutamate receptors is at the center of synaptic plasticity, a process through which adaptive changes in response to environmental stimuli occur in synapses (*Traynelis et al., 2010*). Ionotropic glutamate receptors include the 2-amino-3-(5-methyl-3-oxo-1,2-oxazol-4-yl) propanoic acid (AMPA), N-Methyl-D-aspartic acid (NMDA) and kainite receptors. AMPARs are up- or down-regulated in an experience/activity-dependent way in learning and memory processing (*Granger et al., 2011*; *Liu and Zukin, 2007*). The AMPAR has four subtypes (GluR1 ~4) and acts as a heteromer through combinations of these four subtypes (*Traynelis et al., 2010*). GluR2-lacking AMPARs show inward rectification in the current-voltage (I-V) relationship and exhibit calcium permeability, whereas GluR2-containing AMPARs show a linear expression in an I-V and are calcium impermeable (*Bowie and Mayer, 1995*). GluR2-lacking, calcium-permeable (Cp) AMPARs have a larger conductance than regular GluR2-containing, calcium-impermeable (Ci)-AMPARs and play an important role in the induction of various forms of synaptic plasticity including long-term potentiation (LTP) and long-term depression (LTD) in many neurons (*Isaac et al., 2007*; *Liu and Zukin, 2007*). Increases in the rectification of AMPAR-mediated synaptic currents due to enhanced expression of Cp-AMPARs is a feature of some forms of synaptic plasticity in animals (*Isaac et al., 2007*). Although it is known that glutamate receptors in ARC neurons may be modified during changes in the feeding behaviors of animals (*Horvath, 2006*; *Pinto et al., 2004*), it is not clear how glutamate receptors are regulated in POMC neurons during different metabolic states.

In this study, we examined whether the alteration in POMC circuitry in the ARC occurs at the level of the glutamate receptor in animals under different feeding paradigms.

## Results

### POMC neurons are inhibited by fasting

To investigate excitatory inputs onto the POMC neurons during a negative energy state, miniature excitatory postsynaptic currents (mEPSCs) were recorded by using whole cell voltage clamp in the presence of tetrodotoxin (TTX) (0.5 μM) and picrotoxin (50 μM) in POMC neurons in brain slices from mice fasted overnight (n = 5 mice) and their control counterparts (n = 5 mice). The frequency of mEPSCs was comparable in fasted and fed mice (fasting, 3.65 ± 0.61 Hz, n = 13; feeding, 4.08 ± 0.57 Hz, n = 14, p=0.61, unpaired t-test) (*Figure 1a,b*). The averaged median amplitude of mEPSCs tended to be smaller in fasted mice than in fed controls on normal chow (fasting, 13.46 ± 0.65 pA, n = 13; feeding, 16.37 ± 1.55 pA, n = 14, p=0.10, unpaired t-test) (*Figure 1a,c*). The distribution of amplitude was significantly different between fasted and fed groups (p<0.001, Kolmogorov-Smirnov test) (*Figure 1d*). These data indicate that postsynaptic properties may be altered in POMC neurons in animals undergoing changes in their energy states.

### Cp-AMPARs dominate POMC neurons in fed but not fasted animals

To decipher postsynaptic changes in POMC neurons of animals in different energy states, the AMPAR/NMDAR ratio of evoked excitatory postsynaptic currents (eEPSCs) and rectification of AMPAR-mediated eEPSCs were recorded from fasted and fed animals (on standard chow). The AMPAR/NMDAR ratio was calculated as the ratio of peak amplitude between AMPAR-mediated eEPSC at −60 mV and NMDAR-mediated eEPSC at +40 mV. There was no significant difference between POMC neurons of fasted and fed animals (fasting, 1.672 ± 0.679, n = 6; feeding, 1.415 ± 0.136, n = 6, p=0.73, T-test) (*Figure 2d*). However, the eEPSC recorded in the presence of NMDAR inhibitor, AP5 (50 μM) and GABA$_A$ receptor inhibitor, bicuculline (30 μM) exhibited a significant increase in the rectification of the current-voltage (I-V) relationship of AMPARs in POMC neurons of fed mice compared to fasted animals (*Figure 2a,b*). The rectification of AMPAR-mediated EPSCs is represented by the rectification index (RI), which is calculated as the ratio between eEPSC

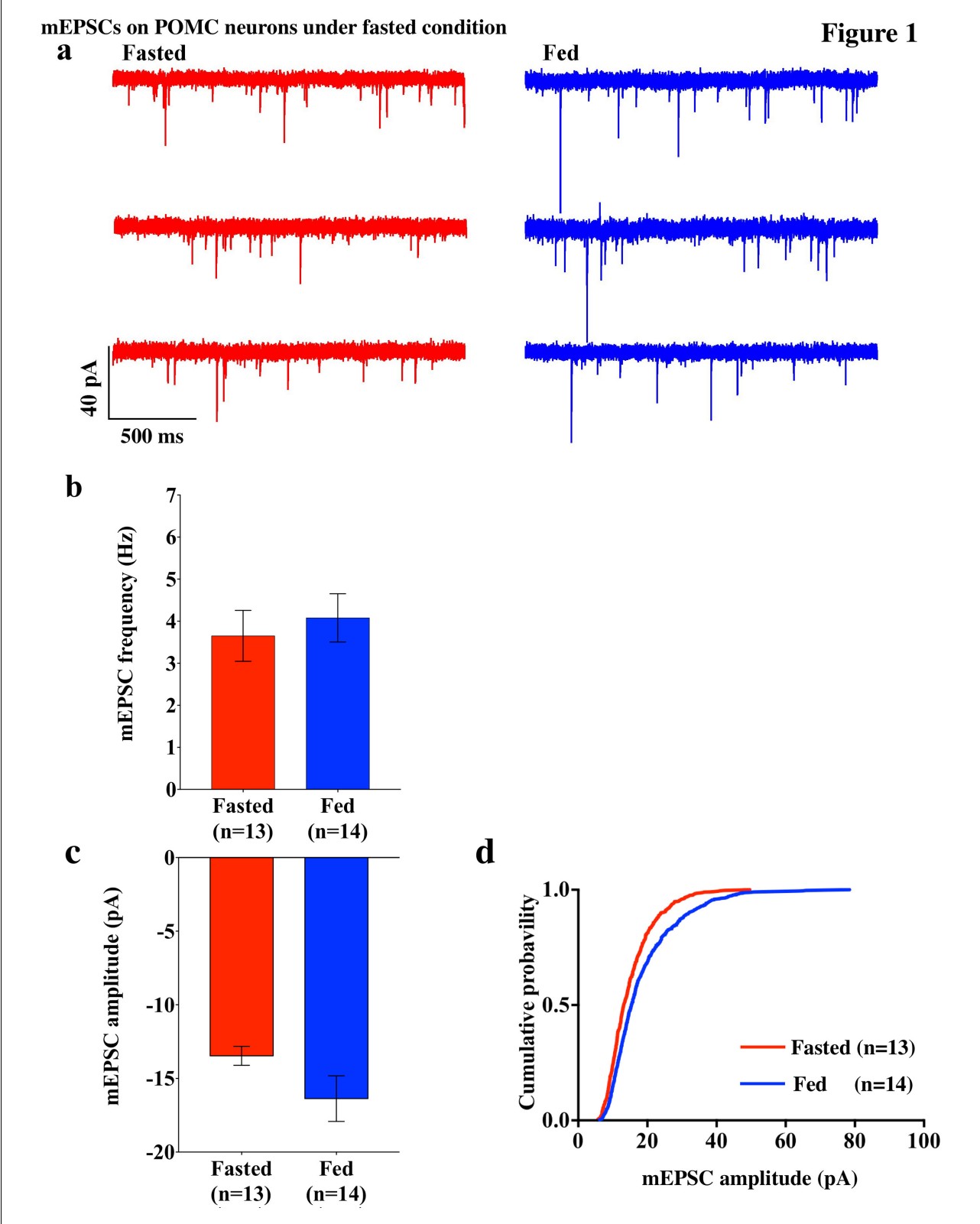

**Figure 1.** mEPSCs on POMC neurons was decreased by fasting. (**a**) Representative trace of mEPSCs on POMC neurons. Scale bar indicates 40 pA on vertical and 500 ms on horizontal. (**b**) mEPSC frequency (mean ± SEM) did not change significantly between overnight fasting (red) and ad lib feeding conditions (blue). (**c**) Averaged median mEPSC amplitude (mean ± SEM) of POMC neurons tended to decrease under fasting (red) than feeding (blue)

*Figure 1 continued on next page*

*Figure 1 continued*

conditions. (d) Cumulative probability of mEPSC amplitude under fasting condition (red) decreased significantly than under feeding condition (blue) (* indicates p<0.01, *Kolmogorov-Smirnov* test).

amplitudes at +40 mV and −60 mV (*Isaac et al., 2007*; *Liu and Cull-Candy, 2000*). The RI of AMPAR-EPSC in POMC neurons was 0.59 ± 0.01 (n = 8) in fasted mice and 0.30 ± 0.03 (n = 7) in fed mice (p<0.05, unpaired t-test) (*Figure 2c*), indicating that the rectification of AMPAR-EPSCs was more significant in fed mice than in fasted ones. This suggests that fasting facilitates GluR2-containing AMPAR translocation to synapses on POMC neurons while feeding leads to an increased expression of GluR2-lacking AMPARs on POMC neurons. Although AMPAR currents should show a reversal potential close to 0 mV, in this experiment, the cells show reversal potentials that are substantially different from 0 mV. The main reason for this error in voltage might be due to a poor space clamp, which is different in individual cells recorded in this study. Therefore, we have re-examined all the recorded cells and excluded some recordings which deviate substantially from 0 mV (possibly representing a poor space clamp) and re-calculated the RIs. In so doing, we found that the RI values were similar to the original results and were still significantly different between fasted and fed groups [0.62 ± 0.11 (n = 6) in fasted mice and 0.32 ± 0.03 (n = 5) in fed mice (p<0.05, unpaired t-test)] (*Figure 2e,f*).

To test whether the change in the composition of AMPARs is selective to POMC neurons during the different mouse feeding paradigms, we examined the AMPAR/NMDAR ratio and rectification of AMPAR-EPSCs in neighboring NPY/AgRP neurons of fasted and fed mice. The AMPAR/NMDAR ratio (fasting, 3.427 ± 0.820, n = 6; feeding, 5.818 ± 1.891, n = 4, p=0.29, unpaired T-test) and RI of AMPAR-EPSCs (fasting, 0.281 ± 0.071, n = 8; feeding, 0.211 ± 0.051, n = 6, p=0.44, unpaired T-test) were comparable in NPY/AgRP neurons between fasted and fed mice (*Figure 3*), which is consistent with the report by *Yang et al. (2011)*.

To verify the increased expression of GluR2-lacking AMPA receptors on POMC neurons in fed but not fasted mice, we examined the effects of a selective open channel antagonist of Cp-AMPAR, IEM 1460, on evoked EPSCs in POMC neurons of fed and fasted mice. The rationale for this experiment is that the inhibition of evoked EPSCs by IEM 1460 should be greater in fed animals than in fasted ones if more GluR2-lacking AMPARs are expressed on the surface of POMC neurons in these fed mice. Evoked EPSCs were recorded in POMC neurons in slices from both fed and fasted mice. After a stable baseline was obtained, IEM 1460 (100 μM) was applied to the recorded neurons through bath application. The normalized eEPSC amplitudes were 84.3 ± 5.6% of baseline (n = 8) in fasted mice and 69.3 ± 5.3% of baseline (n = 7) in fed mice; the inhibition of EPSC amplitude in POMC neurons induced by IEM 1460 was significantly greater in fed than fasted mice (p<0.05, unpaired one-tail T-test) (*Figure 4*). This observation supports the premise that there are more GluR2-lacking AMPARs in POMC neurons in fed versus fasted mice.

Since GluR2-lacking AMPARs are calcium permeable, the removal of extracellular calcium should have an impact on AMPAR-EPSCs in POMC neurons from fed mice. To test this hypothesis, we recorded mEPSCs in POMC neurons in the presence and absence of extracellular calcium ions in both fed and fasted mice. In POMC neurons of fasted mice, the median and distribution of mEPSC amplitude was not significantly different between normal and $Ca^{2+}$-free ACSF (normal ACSF, 13.46 ± 0.65 pA, n = 13; $Ca^{2+}$-free ACSF, 11.64 ± 0.787, n = 9) (*Figures 1a,c* and *5a,b*), while the averaged median and distribution of mEPSC amplitude was significantly decreased in $Ca^{2+}$-free ACSF than in normal ACSF in POMC neurons from fed mice (normal ACSF, 16.37 ± 1.55 pA, n = 14; $Ca^{2+}$-free ACSF, 10.93 ± 0.4866 pA, n = 9, p<0.05 unpaired t-test) (*Figures 1a,c* and *5a,b*). Notably, averaged median and distribution of mEPSC amplitudes without extracellular calcium were not significantly different between POMC neurons of fed and fasted groups [p=0.456 for average mEPSC amplitude (unpaired T-test), p=0.70 for distribution (Kolmogorov-Smirnov test)] (*Figure 5c,d*), indicating that extracellular calcium ions contribute more to the amplitude of mEPSCs in POMC neurons of fed but not fasted mice. This is consistent with GluR2 entry into the synapses in the fasted state.

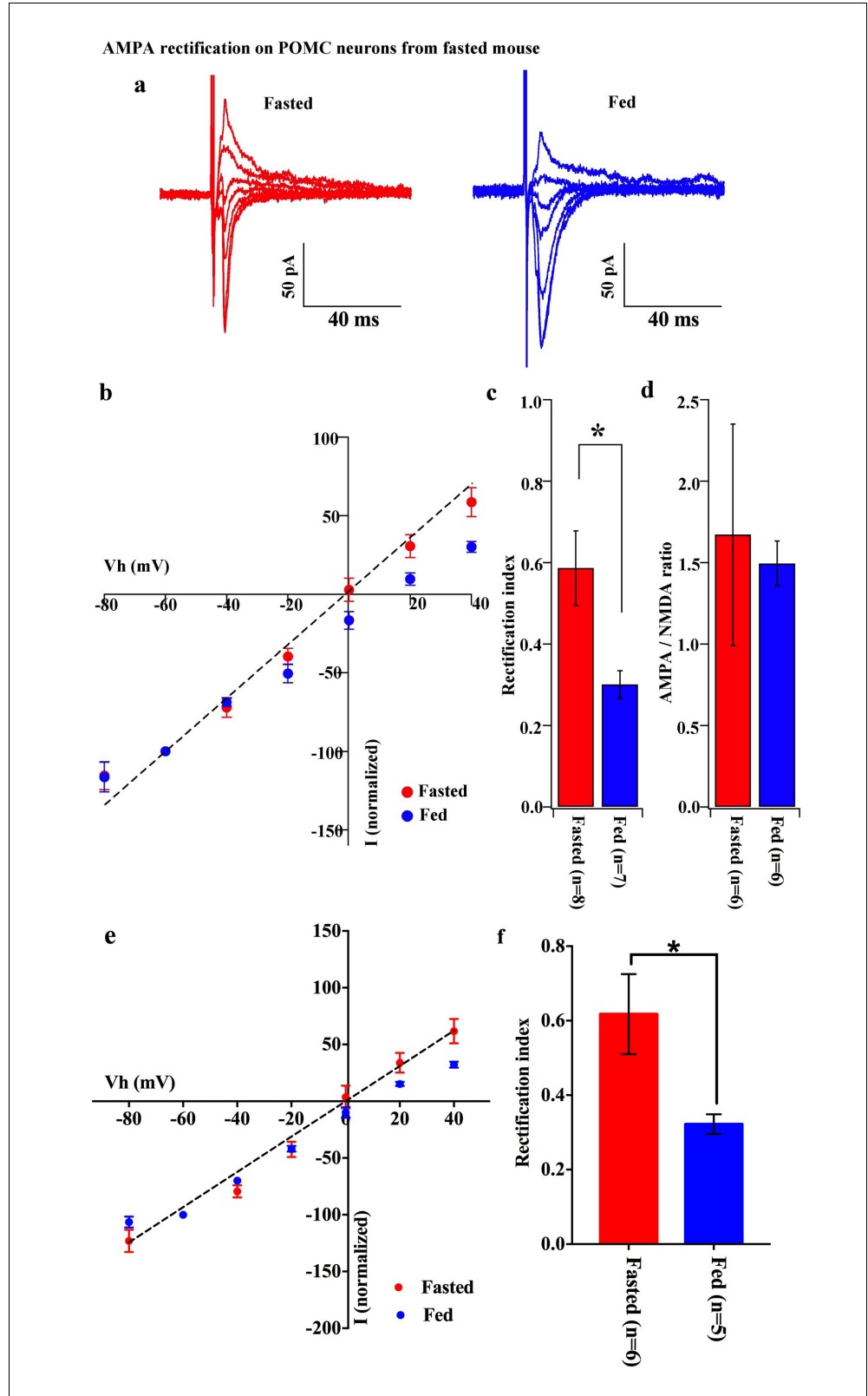

**Figure 2.** AMPA rectification on POMC neurons disappeared but AMPA/NMDA ratio unchanged by fasting. Representative trace of AMPA rectification on POMC neurons under fasting (red) and feeding (blue) conditions. Scale bar indicates 50 pA on vertical and 40 ms on horizontal. (b) I-V relation of AMPARs on POMC neurons under fasted (red) and fed (blue) conditions. The value of peak amplitude of AMPAR-mediated eEPSC was normalized as percent by the value at −60 mV. The dotted line represents a linear regression of AMPAR-EPSCs in POMC neurons between −80 and 0 mV in fasted

*Figure 2 continued on next page*

Figure 2 continued

mice. (c) Rectification index (RI) was calculated by the peak amplitude of AMPAR-EPSC at 40 mV / peak amplitude at −60 mV. RI of fasting condition (red) increased more than feeding condition (blue). (d) AMPA-NMDA ratio did not show a difference between fasting (red) and feeding (blue) conditions. (e) I-V relationship of AMPAR-EPSCs and (f) RI in POMC neurons excluded poor space clamp recordings did not change from original results.

## POMC product secretion needs activation of Cp-AMPAR on POMC neurons

In fed mice, $\beta$ endorphin secretion significantly increased in hypothalamic slices treated with AMPA compared to aCSF, AMPA + the Cp-AMPA blocker IEM 1460 and IEM 1460 (*Figure 6a*). In fasted mice, $\beta$ endorphin levels were comparable in hypothalamic slices treated with AMPA compared to aCSF, AMPA + IEM 1460 and IEM 1460 (*Figure 6b*). This suggests that $\beta$ endorphin could be released via activation of Cp-AMPAR, which is dominant in POMC neurons under fed condition.

## Cp-AMPAR increases on POMC neurons in response to HFD exposure

To test whether positive energy balance leads to altered excitatory transmission via AMPARs in POMC neurons, the rectification of AMPAR-EPSCs in POMC neurons was compared between mice fed a high-fat diet (HFD) and normal diet (ND) (*Figure 7a,b*). HFD was given to mice for 10–14 days to minimize the secondary effects of obesity. The RI was $0.34 \pm 0.04$ (n = 8) in POMC neurons of the ARC of HFD fed mice and $0.46 \pm 0.03$ (n = 9) in ND fed mice. This represents a significant difference in RI (p<0.01, T-test, *Figure 7c*) between these two groups, suggesting that the rectification of AMPAR-EPSCs was significantly enhanced in HFD mice. The RI values excluded poor space clamp recordings were similar to original results and it was still significantly different between HFD and ND groups [$0.33 \pm 0.04$ (n = 7) in fasted mice and $0.48 \pm 0.04$ (n = 7) in fed mice (p<0.05, unpaired t-test), *Figure 7d,e*]. These data indicate that Cp-AMPAR expression at synaptic sites on POMC neurons is enhanced by HFD. To confirm this result, mEPSCs in POMC neurons with or without extracellular calcium were recorded from mice fed HFD or ND. The mEPSC amplitude decreased after removing extracellular calcium in POMC neurons in both HFDs [$Ca^{2+}$ present: $15.91 \pm 2.17$, $Ca^{2+}$, absent: $12.39 \pm 1.33$ pA (mean $\pm$ SEM), p<0.05 for averaged median amplitude (paired T-test), p<0.001 for distribution (Kolmogorov-Smirnov test), n = 8, *Figure 6d,e*] and ND conditions [$Ca^{2+}$ present: $16.06 \pm 1.05$, $Ca^{2+}$ absent: $14.19 \pm 1.27$ pA (mean $\pm$ SEM), p<0.05 for averaged median amplitude (paired T-test), p<0.001 for distribution (Kolmogorov-Smirnov test), n = 8, *Figure 7f,g*]. Interestingly, mEPSC amplitude without $Ca^{2+}$ was smaller in HFD mice compared to ND [p<0.001 for distribution (Kolmogorov-Smirnov test) *Figure 6e*] whereas those with $Ca^{2+}$ did not show a significant difference between the two groups [p=0.56 for distribution (Kolmogorov-Smirnov test) *Figure 7f,g*]. These data suggest that short-term HFD application induces higher calcium permeability through Cp-AMPAR expression on POMC neurons more than ND does.

## Leptin suppresses GluR2 in the AMPA receptor complex of POMC neurons

Both the fed state and HFD feeding are associated with elevated circulating leptin levels. To test the hypothesis that leptin may be involved in the regulation of the AMPA receptor complex, we examined the effects of leptin on the rectification of AMPAR-EPSCs in POMC neurons in the ARC of fasted mice. Saline or leptin (5 mg/kg body weight) was administered to fasted mice by intraperitoneal (IP) injection upon the removal of food. The rectification of AMPAR-EPSCs was examined by measuring the I-V relationship of AMPAR-EPSCs in POMC neurons in hypothalamic slices from fasted and fasted plus leptin mice (*Figure 8a,b*). The RI was $0.489 \pm 0.062$ (n = 8) in fasted plus saline-treated mice, but significantly less in the fasted plus leptin-treated mice: $0.296 \pm 0.057$ (n = 7). (*Figure 8b,c*). The RI values excluded poor space clamp recrdings were similar to original results and it was still significantly different between leptin and saline groups [$0.32 \pm 0.07$ (n = 5) in leptin-treated mice and $0.55 \pm 0.06$ (n = 6) in saline-treated mice (p<0.05, unpaired t-test), *Figure 8f,g*]. To test whether leptin affects ARC POMC neurons directly, we measured the I-V relationship of AMPAR-EPSCs in POMC neurons of slices which were incubated for 4 hr with or without 50 nM leptin, compared to systemic leptin application under the fasted condition. The RI was $0.585 \pm 0.050$

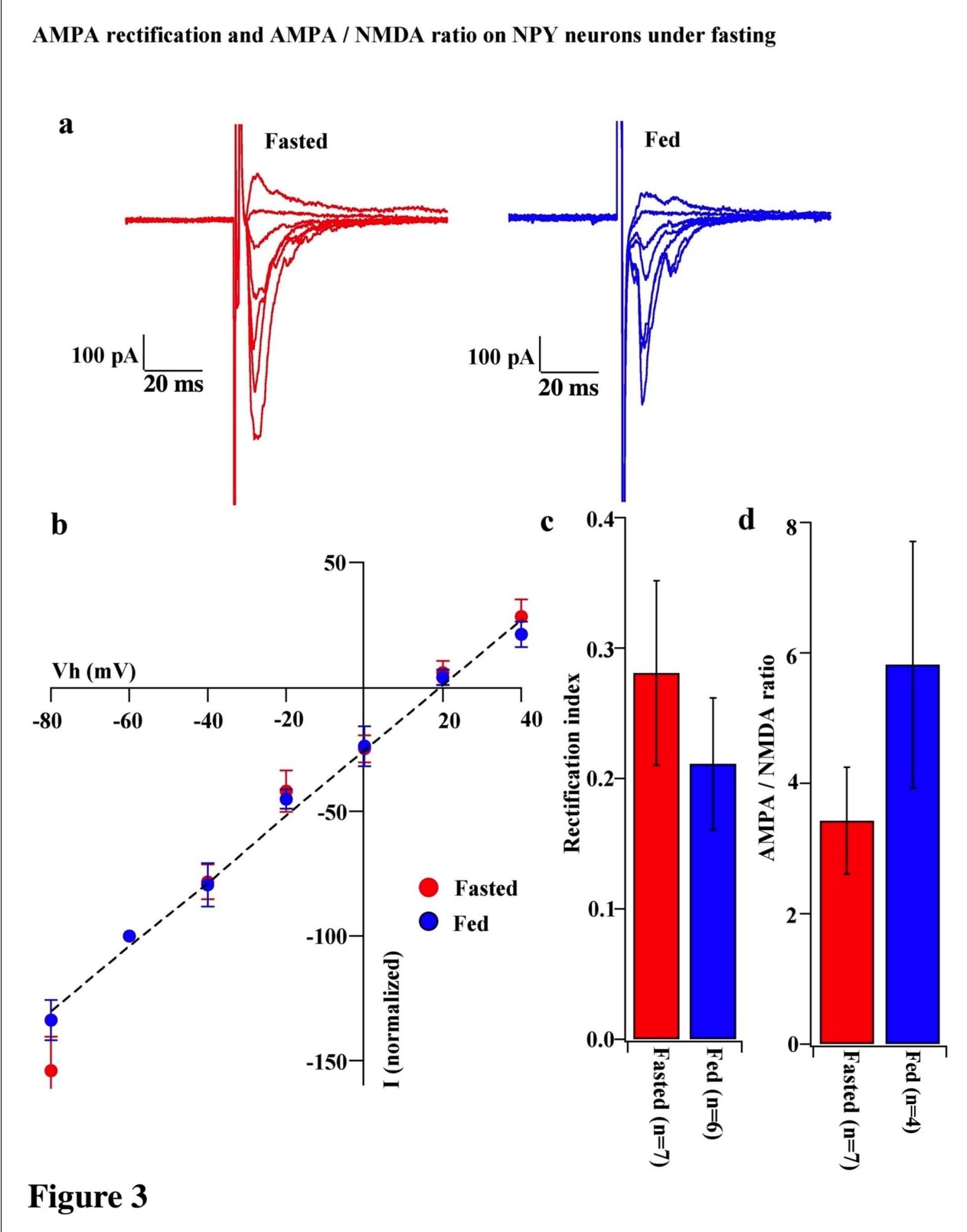

**Figure 3.** AMPA rectification and AMPA / NMDA ratio did not change on NPY neurons during fasting. (a) Representative trace of AMPA rectification in NPY neurons in fasted (red) and fed (blue) mice. Scale bar indicates 100 pA on vertical and 20 ms on horizontal. (b) I-V relationship of AMPAR-EPSCs in POMC neurons under fasting (red) and feeding (blue). The value of peak amplitude of AMPAR-mediated eEPSC was normalized as percent by the value

*Figure 3 continued on next page*

*Figure 3 continued*

at −60 mV. Linear regression (dotted line) represents between −80 and 0 mV under fasting. (**c**) RI of fasted group (red) did not change compared with the fed group (blue). (**d**) AMPA-NMDA ratio did not show a difference between fasted and fed groups.

(n = 3) without leptin treatment and 0.212 ± 0.063 (n = 3) with leptin treatment in slices from fasted mice, thus showing a significantly smaller RI in leptin-treated POMC neurons compared to the control and systemic leptin application (*Figure 8d,e*).

To investigate which downstream pathways of leptin signaling contribute to the regulation of the AMPAR complex, we measured the rectification of AMPAR-EPSCs on POMC neurons under conditions in which mTOR is either inhibited or activated. Saline or rapamycin (inhibitor of mTOR, 10 mg/kg body weight) was given to fed mice by IP injection. Rapamycin treatment caused a significant increase in the RI in POMC neurons compared to saline injected fed mice [rapamycin group: 0.817 ± 0.104 (n = 7), saline group: 0.403 ± 0.066 (n = 7)] (*Figure 9a–c*). The RI values excluded poor space clamp recordings were similar to original results and it was still significantly different between rapamycin and saline groups [0.82 ± 0.12 (n = 6) in rapamycin-treated mice and 0.35 ± 0.05 (n = 6) in saline-treated mice ($p<0.05$, unpaired t-test), *Figure 9d,e*]. Furthermore, the ARC slices from fasted mice incubated with 100 µM leucine (an activator of mTOR) for 4 hr showed a significant decrease in RI [leucine group: 0.301 ± 0.033 (n = 6), control group: 0.498 ± 0.449 (n = 6), $p<0.01$, T-test)] (*Figure 10*). These results provide evidence that the leptin-mTOR pathway contributes to the enhanced expression of Cp-AMPARs in POMC neurons in the state of positive energy balance (fed animals).

## ERK promotes GluR2 in the AMPA receptor complex of POMC neurons

Several lines of evidence suggest that extracellular signal-regulated kinase (ERK) phosphorylation plays a critical role in AMPA translocation in many brain regions (*Aponte et al., 2011*; *Cone, 2005*; *Gao and Horvath, 2007*; *Horvath, 2006*; *Liu et al., 2010*; *Mountjoy et al., 1992*; *Ollmann et al., 1997*; *Schroeter et al., 2007*). To investigate whether ERK signaling also contributes to the regulation of the AMPAR complex on POMC neurons in different energy states, we measured the rectification of AMPAR-EPSCs in POMC neurons from both fed and fasted mice with or without a MEK/ERK-specific inhibitor, PD98095. Ten µM of PD98095 was applied to POMC neurons directly via recording pipettes. The rectification of AMPAR-EPSCs was recorded at 1, 10, and 15 mins after rupture of the cell membrane, with results recorded 1 min after membrane rupture as a control. The RI on the fed group did not show a significant difference between the control at 10 or 15 min after rupture of the membrane. By comparison, the RI on the fasted group decreased significantly at 10 min after rupture [fasted group: 0.60 ± 0.098 at 1 min, 0.353 ± 0.038 at 10 min, 0.344 ± 0.048 at 15 min, (n = 7) $p<0.01$ in 1 min vs. 10 min and 1 min vs. 15 min (two-way analysis of variance (ANOVA) and Tukey multiple comparison posthoc test), fed group: 0.426 ± 0.064 at 1 min, 0.409 ± 0.068 at 10 min, 0.332 ± 0.051 at 15 min (n = 7)] (*Figure 11a–c*). The RI values excluded poor space clamp recrdings were similar to original results and it was still significantly different in fasted but not in fed groups [fasted group: 0.68 ± 0.12 at 1 min, 0.39 ± 0.04 at 10 min, 0.34 ± 0.06 at 15 min (n = 5), $p<0.01$ in 1 min vs. 10 min and 1 min vs. 15 min (two-way analysis of variance (ANOVA) and Tukey multiple comparison posthoc test), fed group: 0.32 ± 0.01 at 1 min, 0.36 ± 0.02 at 10 min, 0.27 ± 0.01 at 15 min (n = 4), *Figure 11d–f*]. These data suggest that ERK phosphorylation promotes GluR2-containing Ci-AMPAR translocation to the synaptic membrane.

## Discussion

In this study, we demonstrated that excitatory inputs onto POMC neurons in the ARC facilitate synaptic changes that are dependent on the metabolic state. The amplitude of mEPSCs on POMC neurons from fasted mice is significantly smaller than in the fed group, whereas the frequency of mEPSCs is comparable between POMC neurons in fasted and fed mice. Enhancement in mEPSC amplitude is likely due to the changes in the composition of glutamate receptor subunits in POMC neurons. Our data showed that the rectification of AMPAR-mediated EPSC is significantly increased in mice fed with standard chow as compared to food-restricted mice. This is consistent with the effect of positive energy balance on these processes: HFD leads to further rectification of AMPAR-

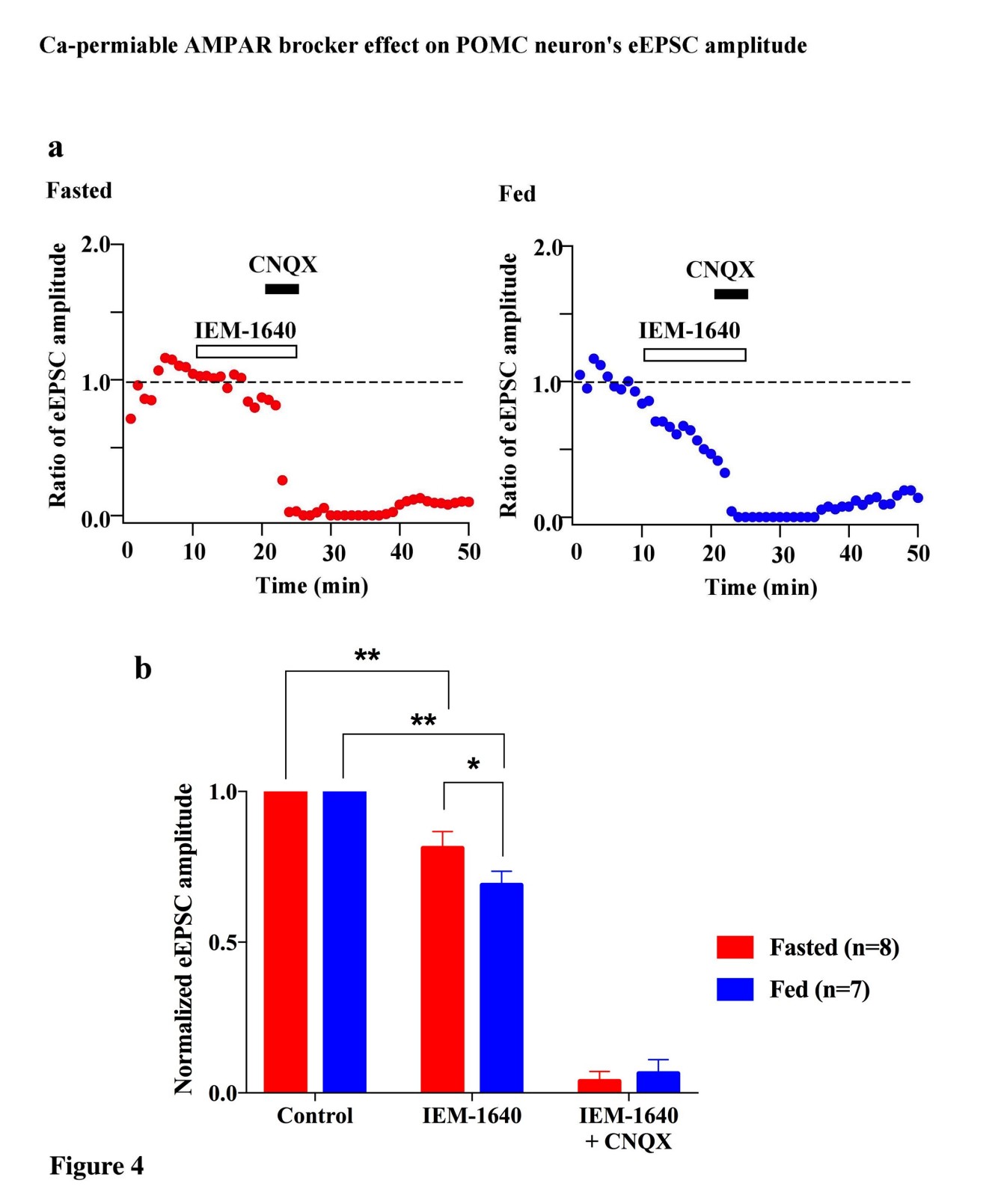

**Figure 4.** Cp-AMPARs blocker, IEM-1460, inhibited evoked EPSCs under feeding but not fasting. (a) Representative trace of normalized eEPSC amplitude under fasting (red) and feeding (blue). 100 µM IEM-1640, competitive Cp-AMPARs blocker, was applied to bath solution at 10–25 min. 10 µM CNQX, AMPARs blocker, was also applied to bath solution at 20–25 min. (b) Average normalized eEPSC amplitude onto POMC neurons under fed condition was decreased by IEM 1460 but not under fasted condition.

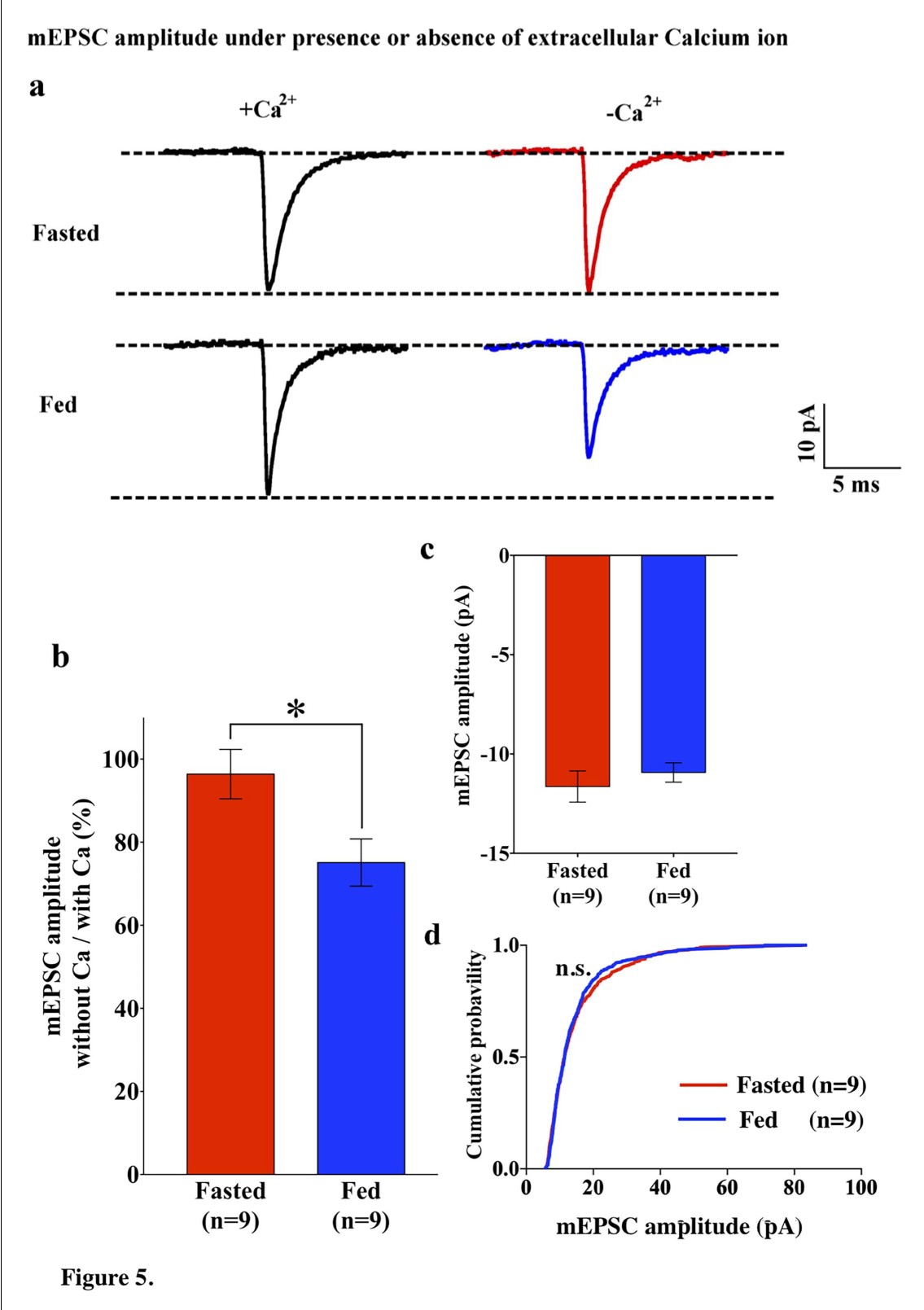

**mEPSC amplitude under presence or absence of extracellular Calcium ion**

**Figure 5.**

**Figure 5.** mEPSC without extracellular calcium did not change under fasting, whereas it decreased under feeding conditions. (a) Representative traces of mEPSC with calcium (left) and without calcium (right) under fasting (top) and feeding (bottom) conditions. Scale bar indicates 10 pA on vertical and 5 ms on horizontal. (b) The ratio of mEPSC amplitude without calcium/with calcium decreased under the feeding (blue) but remained unchanged under

*Figure 5 continued on next page*

*Figure 5 continued*

fasting (red). (**c**) mEPSC distribution did not change between fasting and feeding (blue) in $Ca^{2+}$-free ACSF (p>0.05, *Kolmogorov-Smirnov* test) (**d**) Averaged median mEPSC amplitude did not change between fasting and feeding in $Ca^{2+}$-free ACSF.

EPSCs in POMC neurons as compared with mice under standard chow, and fasting-induced altered rectification of AMPAR-EPSCs on POMC neurons was prevented by leptin via the mTOR pathway. These results indicate that the expression of GluR2-lacking calcium-permeable AMPARs on the surface of POMC neurons in the ARC is inherent in the adaptation of POMC neuronal activity when animals are fed or in positive energy balance. The use of a selective antagonist for the GluR2-lacking receptor, IEM1460, and $Ca^{2+}$-free ACSF, verified the appearance of Cp-AMPARs on POMC neurons. The changes in the composition of glutamate receptor subunits according to the feeding paradigm or energy state in animals are specific to POMC neurons, since in NPY/AgRP neurons in the ARC, AMPAR-mediated synaptic currents do not show a difference in the rectification between fed and fasted mice.

Although AMPAR currents should show a reversal potential close to 0 mV, in this study the cells show reversal potentials that are substantially different from 0 mV. The errors in voltage reported in this manuscript could originate from liquid junction potential (LJP), series resistance (Rs), opening of $K^+$ channels at positive membrane potential, and poor space clamp, etc. Many of these factors could be compensated for (such as LJP, Rs and opening of K+ channels) but the poor space clamp is, at present, a limitation in electrophysiological slice preparations. We have compensated for the LJP and Rs; also, we used $Cs^+$ instead of $K^+$ in the pipette solution (please see Materials and methods). The main reason for the error in voltage described in this manuscript is probably due to a poor space clamp, which is different in individual cells recorded in this study. Therefore, we have re-examined all the recorded cells and excluded some recordings in which the reversal potentials deviated substantially from 0 mV and re-calculated the RIs. As a result, the RI values were similar to the original results and were still significantly different between control and test groups as they were in the original results. Therefore, since the space clamp problem existed equally in all measurements among all groups, our original conclusions were not affected by this issue.

## Role of Cp-and Ci-AMPARs on POMC neurons

According to a widely accepted model of the role of the ARC in the regulation of energy homeostasis, NPY/AgRP neurons in this region are responsible for negative energy balance, while POMC neurons are responsible for positive energy balance (*Acuna-Goycolea and van den Pol, 2009*; *Gao et al., 2007*; *Horvath, 2006*; *Horvath et al., 2010*; *Horvath and Gao, 2005*; *Pinto et al., 2004*; *Sotonyi et al., 2010*). Under food restriction, the NPY/AgRP neurons are activated and synaptic plasticity occurs in these cells (*Abizaid et al., 2006*; *Andrews et al., 2008*; *Dietrich et al., 2010*; *Dietrich and Horvath, 2011*; *Granger et al., 2011*; *Liu and Zukin, 2007*; *Liu et al., 2012*), whereas POMC neurons are inhibited and the levels of POMC mRNA are decreased (*Korner et al., 1999*; *Mizuno and Mobbs, 1999*; *Sternson et al., 2005*; *Traynelis et al., 2010*). In this study, we showed that mEPSC amplitude which reflects efficiency of the synaptic transmission at the postsynaptic site is higher in POMC neurons in the ARC in fed animals than in fasted animals, suggesting that under a normal feeding paradigm a higher efficiency of glutamatergic transmission onto POMC neurons may lead to the activation of POMC neurons. Alternatively, it can also be interpreted as a depression of efficiency of glutamatergic transmission onto POMC neurons during food restriction. Regardless, the changes in the composition of glutamate receptors onto POMC neurons under different energy states are apparent. Firstly, in calcium-free ACSF, the amplitude of mEPSCs in the fasting group did not decrease, whereas mEPSC amplitude in the feeding group did. Secondly, the fact that the amplitudes from feeding and fasting groups were comparable under calcium-free conditions (*Figure 5c,d*) indicates that the higher amplitude of mEPSCs in fed mice is due to the calcium influx via Cp-AMPARs. Lastly, a selective Cp-AMPAR blocker, IEM 1460, decreased eEPSC amplitude in fed mice but not in the fasted group, indicating a contribution of Cp-AMPARs to mEPSCs in fed animals. Therefore, our results demonstrate that more GluR2-lacking, $Ca^{2+}$-permeable AMPARs are expressed on the surface of POMC neurons in animals under normal feeding and HFD. This is

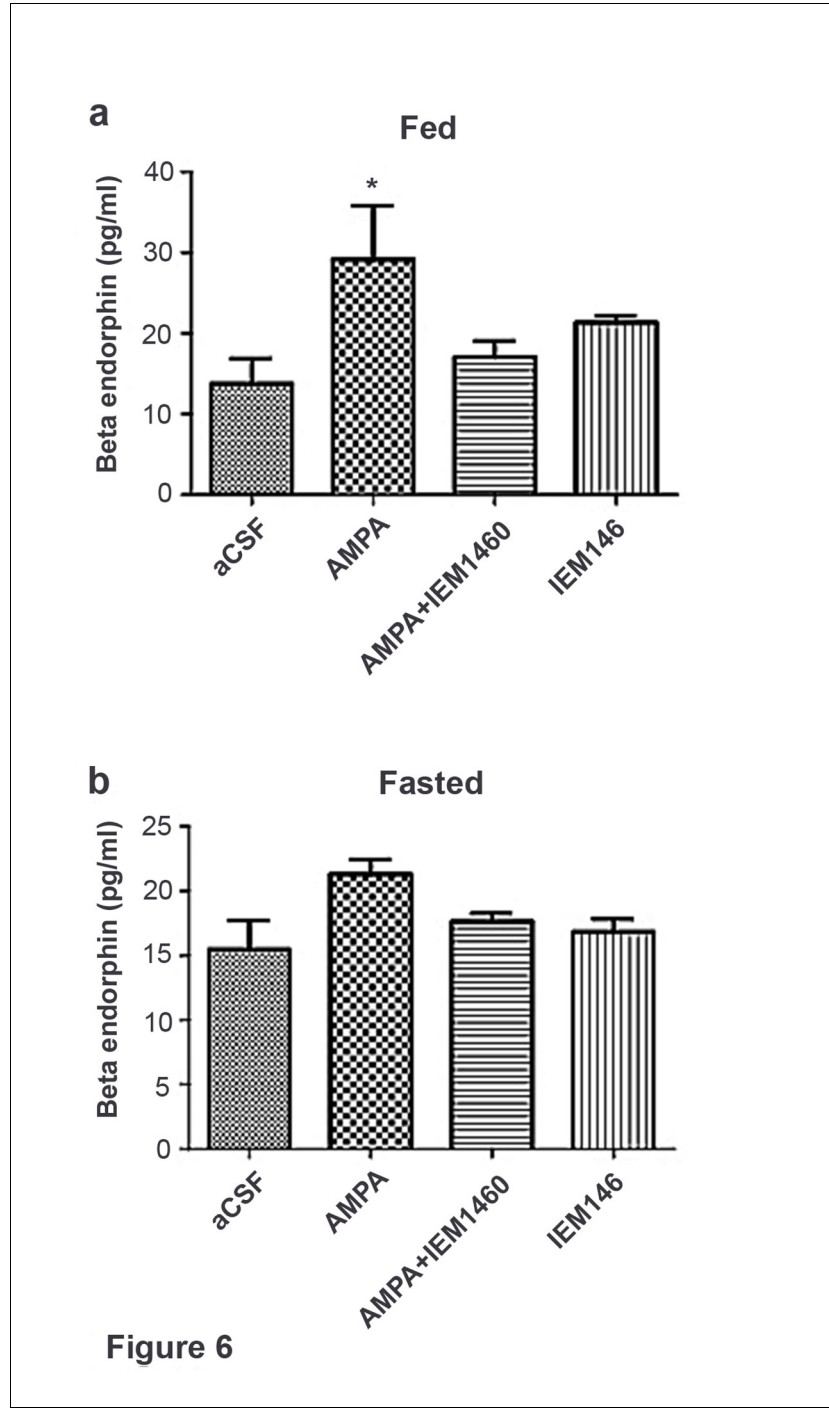

**Figure 6.** Beta-endorphin secreted by AMPA application was inhibited by Cp-AMPAR blocker, IEM 1460 under fed but not under fasted conditions. (a) Under fed condition, beta-endorphin secretion from hypothalamus slice was significantly increased by AMPA application compared to aCSF and AMPA + IEM 1460. (b) Under fasted condition, beta-endorphin secretion was not increased by AMPA, aCSF and AMPA + IEM 1460 treatment. Results are expressed mean ± SEM. * indicates p<0.05, two-way analysis of variance (ANOVA) and Tukey multiple comparison posthoc test.

consistent with a previous study showing that GluR1 but not GluR2 is expressed in ARC neurons under a normal feeding paradigm (*Bowie and Mayer, 1995*; *Gao et al., 2007*).

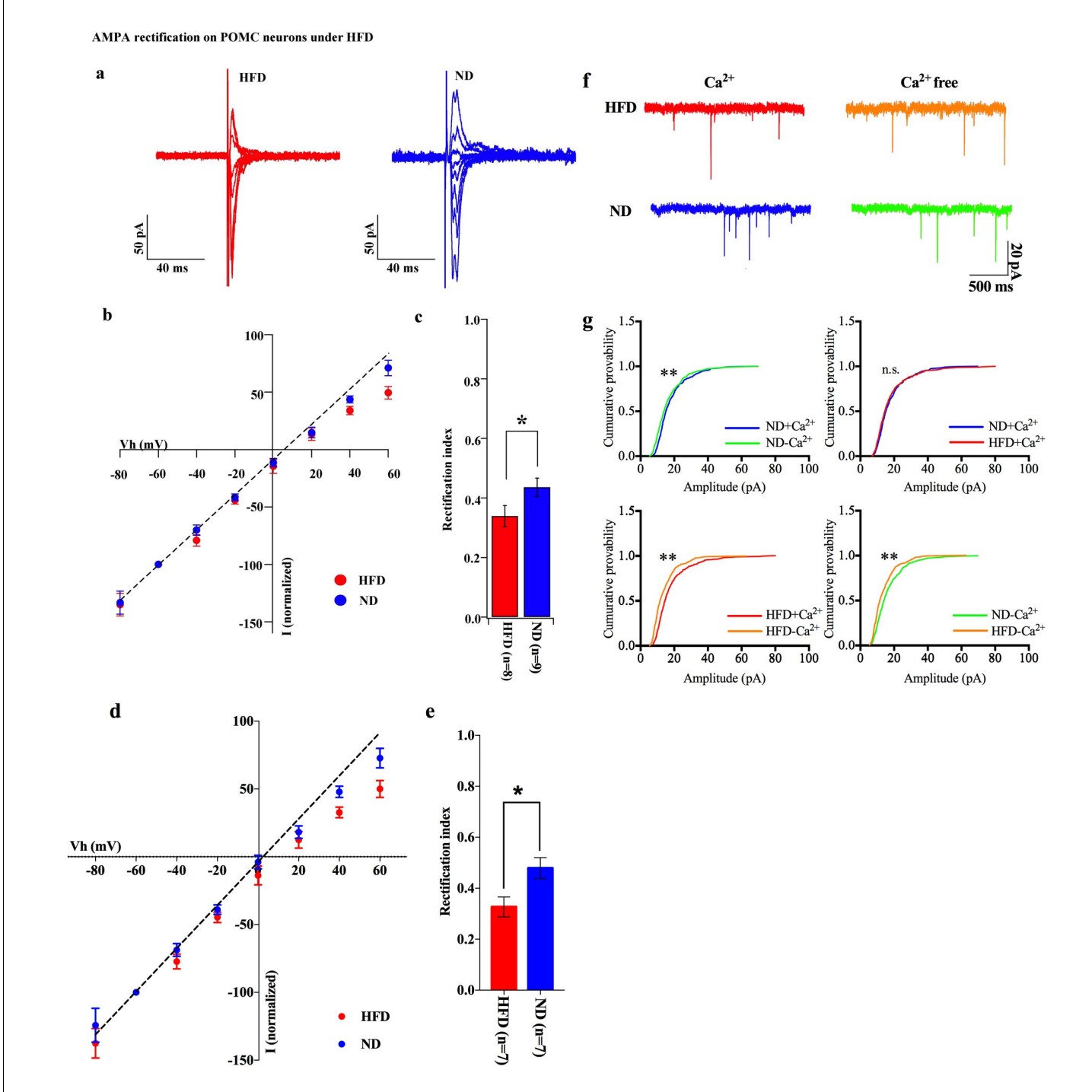

**Figure 7.** HFD increases AMPA rectification in POMC neurons. Representative traces of AMPA rectification under HFD (red) and ND (blue) conditions.
(b) I-V relationship of AMPAR-EPSCs in POMC neurons under HFD (red) and ND (blue). The value of peak amplitude of AMPAR-mediated eEPSC was normalized as percent by the value at −60 mV. The dotted line represents a linear regression of AMPAR-EPSCs in POMC neurons between −80 and 0 mV. (c) The RI of HFD (red) decreased less than ND (blue) significantly (mean ± SEM. * indicates p<0.05, unpaired t-test).) (d) I-V relationship of AMPAR-EPSCs and (e) RI in POMC neurons excluded poor space clamp recordings did not change from original results. (f) Representative trace of mEPSC with or without extracellular calcium under HFD (red) and ND (blue). (g) The distribution of mEPSC amplitude of ND vs. ND without calcium (upper left), ND vs. HFD with calcium (upper right), HFD with or without calcium (lower left), ND vs. HFD without calcium (lower right). (* and ** indicates p<0.05 and p<0.01, respectively, *Kolmogorov-Smirnov* test.

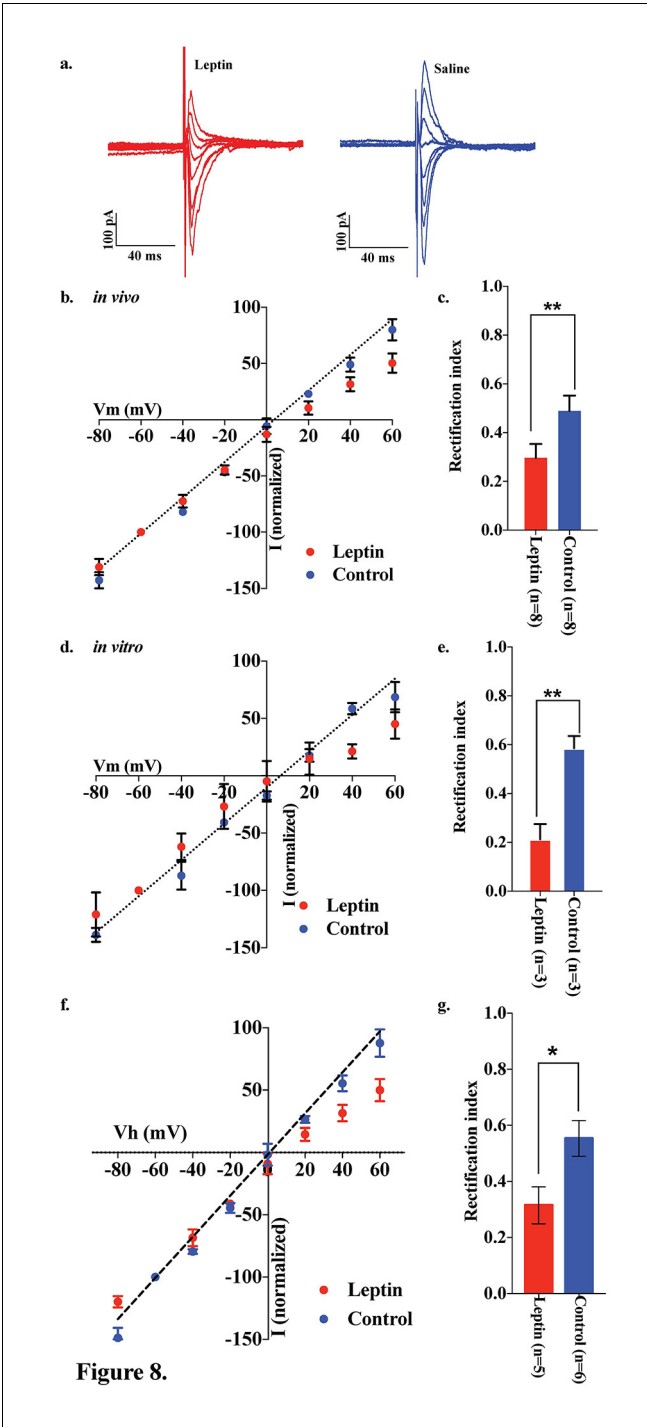

**Figure 8.** Leptin reversed the elimination of AMPAR rectification in POMC neurons in fasted mice. Representative trace of AMPA rectification with (red) or without (blue) IP leptin injection (5 mg/kg body weight) in fasted mice. (**b**) I-V relationship of AMPARs in POMC neurons after leptin (red) or saline (blue) injection (i.p.) in fasted mice. The value of peak amplitude of AMPAR-mediated eEPSC was normalized as percent by the value at −60 mV. Liner regression (dotted line) represent between −80 and 0 mV under control group. (**c**) RIs of AMPAR-EPSCs in POMC neurons in ARC slices in fasted mice treated with 50 nM leptin (red) or saline (blue) in vivo. (**d**) I-V relationship of AMPARs in POMC neurons with (red) or without (blue) leptin treatment to acute brain slices including ARC in fasted mice. (**e**) RIs of AMPAR-EPSCs in POMC neurons in ARC slices from fasted mice treated with leptin (red) or ASCF (blue) in vitro. (mean ± SEM. ** indicates p<0.01, paired t-test.) (**f**) I-V relationship of AMPAR-EPSCs and (**g**) RI in POMC neurons excluded poor space clamp recordings did not change from original results.

There is ample evidence that GluR2 is not expressed in certain types of neurons in many brain areas, including neocortical, hippocampal, and amygdaloidal fast-spiking interneurons, cerebellar stellate cells, dorsal horn interneurons, large striatal cholinergic neurons, bushy and stellate cells of the cochlear nucleus, spinothalamic projection neurons, and retinal amacrine cells, in addition to Bergmann glia and oligodendrocyte precursor cells of the cerebellum under basal conditions. Although most principal neurons of neocortex, hippocampus, cerebellum cortex, and amygdala express GluR2-containing AMPA receptors, some of these neurons show NMDAR-dependent synaptic plasticity. These GluR2-lacking neurons have similar morphological characteristics, for example possessing aspiny dendrites (*Goldberg et al., 2003*; *Isaac et al., 2007*; *Kullmann and Lamsa, 2011*; *Liu and Zukin, 2007*). Studies showed that dendrites of POMC neurons were aspiny (*Hentges et al., 2005*; *Isaac et al., 2007*; *Liu et al., 2012*), which is consistent with previous reports on neurons expressing GluR2-lacking AMPARs. Moreover, some of these neurons were revealed to express NMDAR-independent synaptic plasticity. For example, cerebellar stellate cells express Cp-AMPAR-dependent and NMDAR-independent LTD after high-frequency stimulation, which includes the replacement of Cp-AMPARs by $Ca^{2+}$-impermeable GluR2-containing AMPARs after the expression of LTD (*Liu and Cull-Candy, 2000*). It is important to note that in a recent study, the depletion of NMDARs specifically in POMC neurons did not have a significant effect on feeding and energy expenditure in animals, suggesting that NMDARs may not be required for the functions exerted by POMC neurons (*Liu et al., 2012*).

Additionally, we showed that beta-endorphin secretion relies on activation of Cp-AMPAR (*Figure 6*). Cp-AMPAR may act as Ca2+ source to trigger dendritic release of POMC products. The POMC gene encodes anorexigenic peptide alpha-MSH and opioid peptide beta-endorphin which act as orexigenic signal (*Dube et al., 1994*; *Kalra and Horvath, 1998*). We have reported that activation of cannabinoid receptor in POMC neurons promotes beta-endorphin release but not alpha-MSH and that this process enables increasing food intake evoked by cannabinoids at the state of satiety (*Koch et al., 2015*). However, it remains unknown whether activation of Cp-AMPAR promotes the secretion beta-endorphin evoked by v=cannabinoids.

## The role of leptin in re-composition of AMPARs in POMC neurons

In our experiments, the AMPA-EPSC rectification was measured from mice on HFD for 10–14 days. Under these conditions, it is expected that the secondary effects originating from obesity by HFD application were minimized, because HFD did not induce weight change until at least 2 weeks after starting it (*Lin et al., 2000*). Circulating levels of leptin have been reported to increase at 3 days after HFD application (*Wang et al., 2001*). In this period, POMC neurons were rewired in a polysialic acid molecule-dependent fashion, and showed increasing frequency of EPSCs but no change in amplitude of PSCs (*Benani et al., 2012*). Consistent with this report, we observed that the amplitude of mEPSCs did not change in POMC neurons from HFD and ND mice with extracellular calcium. Under calcium-free conditions, however, amplitude of mEPSCs from HFD mice decreased less than those from ND mice, indicating that the Cp-AMPAR-derived synaptic current is larger in mice fed HFD than in mice fed ND, although the comparable positive electric charges go through AMPARs in both situations. These data are consistent with our results of eEPSC-AMPAR rectification. Although total synaptic transmission did not change between HFD and ND, calcium influx may be overloaded in POMC neurons in HFD mice during adaption to the changes accompanied by positive energy balance.

Under fasting conditions, the levels of circulating leptin decline, leading to a reduction in the number and efficacy of excitatory inputs onto POMC neurons (*Ahima et al., 1996*; *Pinto et al., 2004*). Our result showing that replacement of systemic and local leptin in fasted mice increased the rectification of AMPAR-EPSCs in POMC neurons (*Figure 8*) is consistent with these previous reports. Thus, leptin promotes the expression of GluR2-lacking $Ca^{2+}$-permeable AMPARs at the postsynaptic site. We also observed that rapamycin-mediated inhibition and leucine-mediated activation of mTOR decreased while simultaneously increasing the rectification of AMPAR-EPSCs on POMC neurons in fed and fasted mice, respectively (*Figures 9* and *10*). mTOR was shown to be downstream of the leptin signal via PI3K pathway, in order to suppress food intake and regulate POMC expression in the hypothalamus (*Blouet et al., 2009*; *Cota et al., 2006*). There is also evidence that leptin directly regulates Cp-AMPAR translocation by increasing GluR1 expression at synapses of hippocampal CA1 neurons via PTEN inhibition and activation of the PI3K pathway (*Moult et al., 2010*). Leptin also

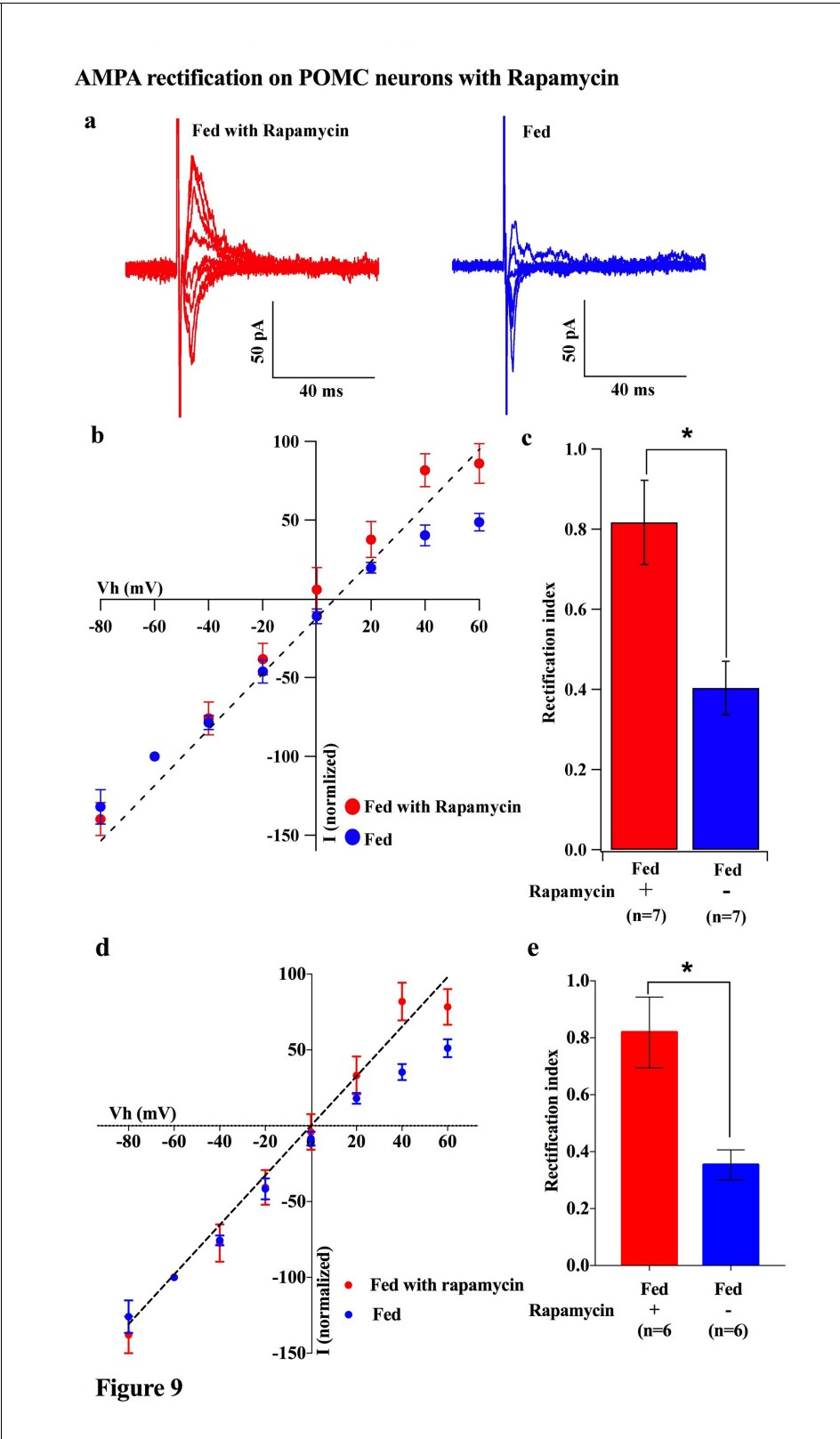

**Figure 9.** Rapamycin blocked AMPAR rectification in POMC neurons in fed mice. (**a**) Representative trace of AMPA rectification with (red) or without (blue) IP rapamycin injection (10 mg/kg body weight) under fed conditions. (**b**) I-V relationship of AMPARs in POMC neurons in fed mice treated with rapamycin (red) or saline (blue). The value of peak amplitude of AMPAR-mediated eEPSC was normalized as percent by the value at −60 mV. The dotted line represents a linear regression of AMPAR-EPSCs in POMC neurons between −80 and 0 mV in control group. (**c**) RIs of AMPAR-EPSCs in

Figure 9 continued

POMC neurons in fed mice treated with rapamycin (red) or saline (blue). (d) I-V relationship of AMPAR-EPSCs and (e) RI in POMC neurons excluded poor space clamp recordings did not change from original results (mean ± SEM. * indicates p<0.05, paired t-test.).

removed Cp-AMPARs after LTP induction via calcium/calmodulin-dependent phosphatase and calcinurin at certain time points after induction of LTP (*Moult et al., 2009*). It will be of importance to examine whether these pathways play a role in facilitating changes in glutamate receptor compositions in POMC neurons. Additionally, growing evidence has shown that mTOR-mediated protein translation is needed to establish several types of long-lasting synaptic changes and long-term memory (see review [*Costa-Mattioli et al., 2009*]). For example, brain-derived neurotrophic factor (BDNF), which is essential for LTP, induced GluR1 translation that required mTOR activation in hippocampal CA1 neurons (*Fortin et al., 2012*; *Slipczuk et al., 2009*). Furthermore, a recent study suggested that mTOR-mediated autophagy control of the GluR1 subunit (*Shehata et al., 2012*). These data, taken together with a recent report that autophagy in POMC neurons maintained neurite growth (*Coupé et al., 2012*), indicates that mTOR may regulate AMPAR turnover at the postsynaptic site of POMC neurons to regulate feeding activity.

Fasting not only induces a decline in leptin but also increases glucocorticoid (*Champy et al., 2004*) which also regulates mTOR (*Polman et al., 2012*). Glucocorticoid decreases POMC mRNA expression, and moreover, it regulates synaptic plasticity in the hippocampus (*Popoli et al., 2011*). Therefore, glucocorticoid is a candidate regulator of AMPARs composition in POMC neurons during fasting. However, a decline in glucocorticoid after adrenalectomy did not alter the amplitude of mEPSCs onto POMC neurons, although it depolarized POMC neurons and increased the frequency of mEPSCs (*Gyengesi et al., 2010*), suggesting that it does not likely regulate translocation of AMPARs in POMC neurons.

We also observed that inhibition of ERK decreased the RI increased by fasting, which indicates that ERK may promote Ci-AMPAR translocation to the synaptic membrane of POMC neurons during the fasted but not fed states. This is consistent with the role of ERK to switch the AMPAR phenotype from Cp-AMPA to Ci-AMPA in several neurons including hippocampus, cortex (*Schroeter et al., 2007*), and cerebellar stellate cells (*Liu et al., 2010*). Our finding is also consistent with the observation that ERK phosphorylation was induced under the fasted but not fed state, and this phosphorylation was reversed by re-feeding (*Morikawa et al., 2004*; *Ueyama et al., 2004*). Although several lines of evidence suggest that leptin induces ERK phosphorylation in the ARC (*Bjørbaek et al., 2001*; *Bouret et al., 2012*), triggers of ERK activation are not only known in the leptin pathway but also in the $Ca^{2+}$-stimulated Ras activation pathway, PKC pathway, and G-protein-coupled receptor pathway (*Peng et al., 2010*). Under the fasted condition, it is thought that other signaling pathways are overridden to activate ERK against the leptin pathway.

Our results suggest that POMC neurons consist of two subpopulations; approximately half of these neurons showed high RI values (greater than 0.7) during fasting conditions, while the others displayed lower RI values within the range of 0.25–0.5 which is similar to POMC neurons under a fed condition. POMC neurons with high RI under fasting conditions could be under the regulation of the leptin-mTOR pathway, whereas those with low RI could be leptin insensitive. This result is consistent with the evidence that a subset of POMC neurons respond to leptin but not serotonin while the other subset responds in the opposite way (*Gao et al., 2017*; *Sohn et al., 2011*)

In summary, the results of our study reveal a novel mechanism in which different energy states of animals contribute to the modification of glutamatergic synapses in central neurons. The modulation of glutamate receptor structure has been shown to be a key part of neuronal responses to changes in the external environment, including learning, memory, and drug addition. Because our observations emphasize the importance of calcium entry into POMC neurons during their activation, our results provide a feasible mechanistic explanation for long-term impairment of POMC neuronal functions on HFD in relation to calcium overload (*Paeger et al., 2017*).

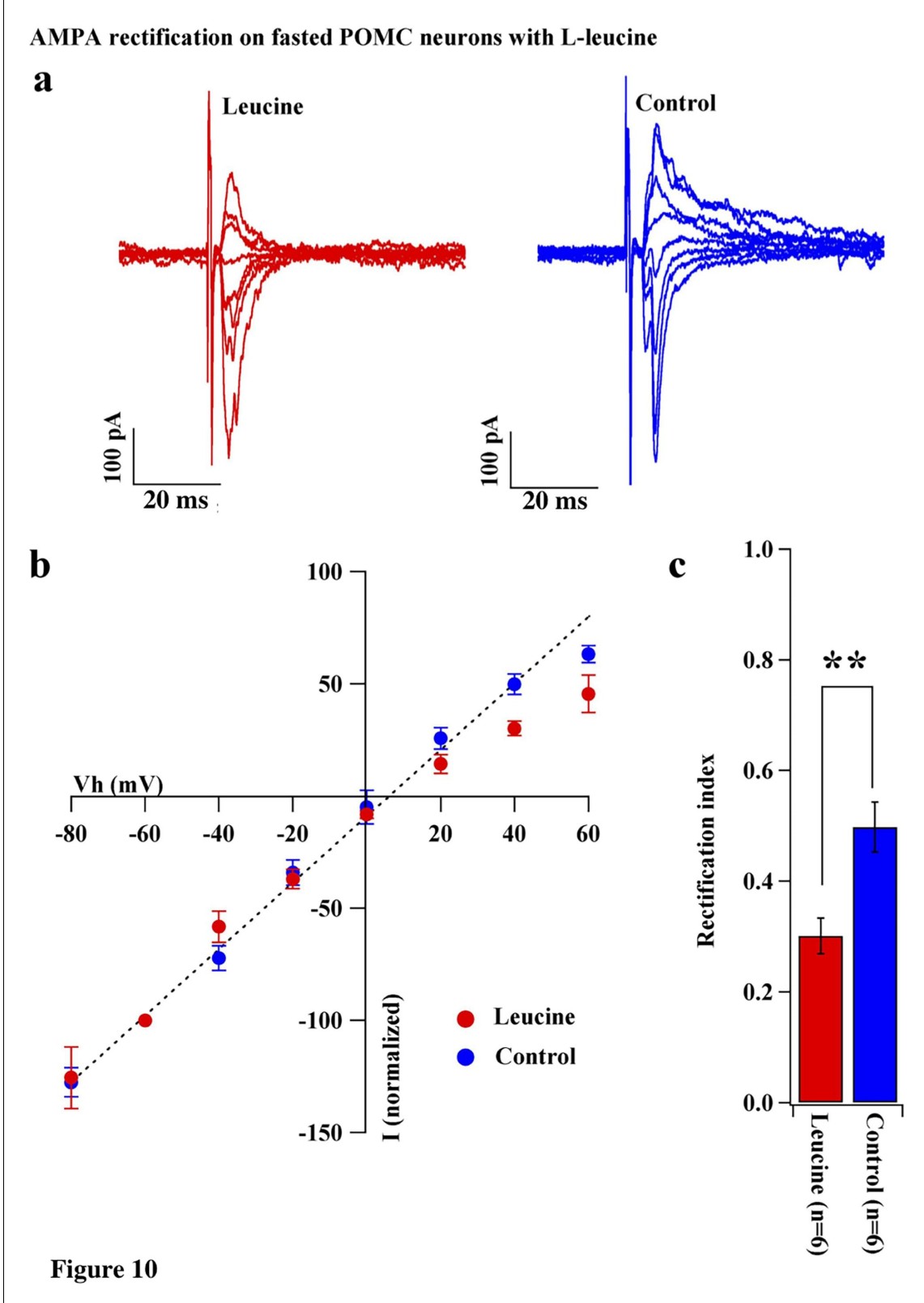

**Figure 10.** Leucine promoted AMPAR rectification in fasted mice. (a) Representative trace of AMPA rectification with (red) or without (blue) leucine (100 μM) application to acute brain slice including ARC in fasted mice. (b) I-V relationship of AMPAR-EPSCs in POMC neurons in fasted mice treated with leucine (red) or saline (blue) in vitro. The value of peak amplitude of AMPAR-mediated eEPSC was normalized as percent by the value at −60 mV. The *Figure 10 continued on next page*

*Figure 10 continued*

dotted line represents a linear regression of AMPAR-EPSCs in POMC neurons between −80 and 0 mV in fasted mice. (**c**) RIs of AMPAR-EPSCs in POMC neurons in fasted mice treated with leucine (red) or saline (blue). (mean ± SEM. * indicates p<0.05, paired t-test.).

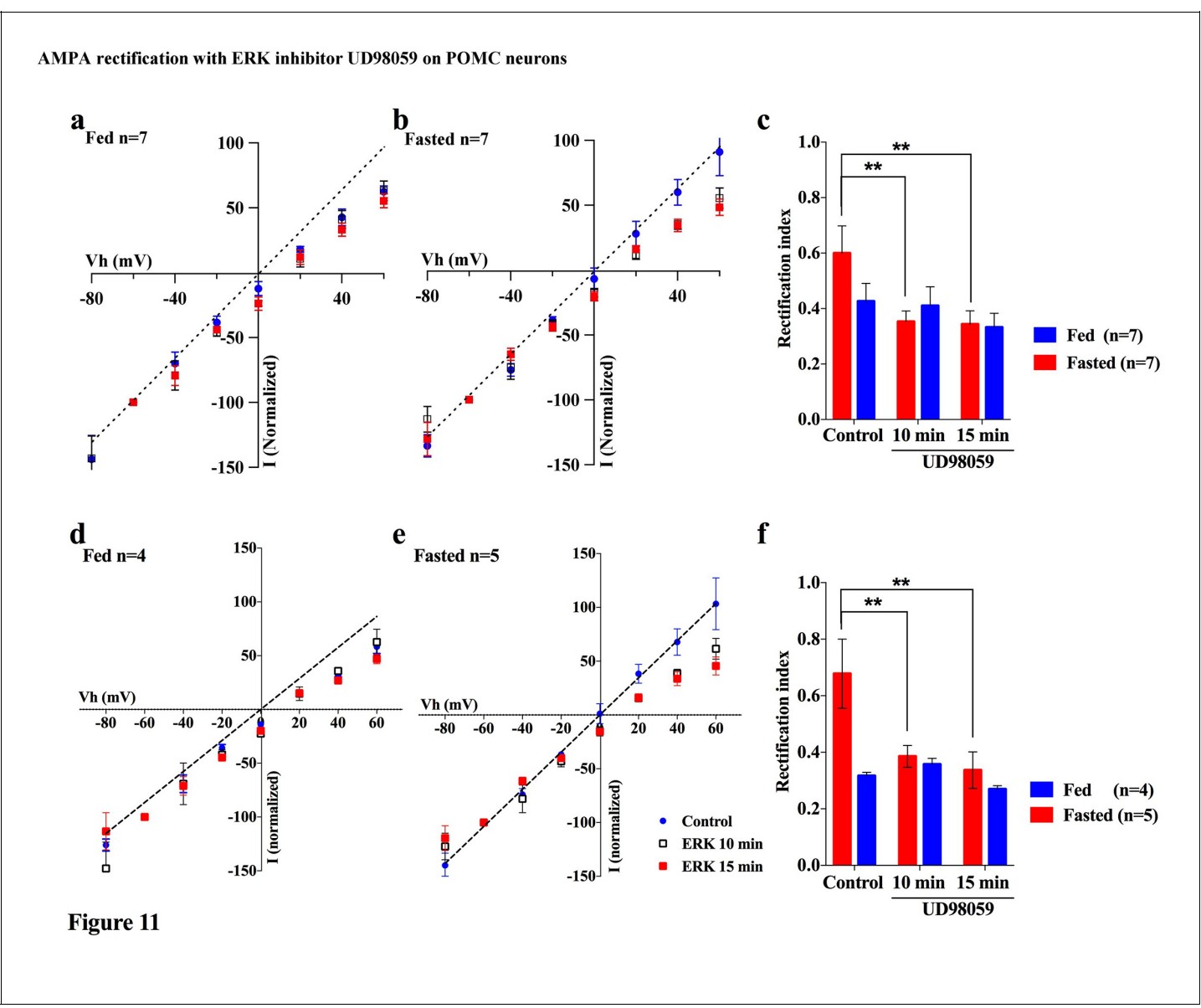

**Figure 11.** An ERK inhibitor promoted AMPAR rectification in fasted but not fed mice. (**a**) I-V relationship of AMPAR-EPSCs in POMC neurons in fed mice at 0 (blue), 10 (box), and 15 (red) min after application (in vitro) of ERK inhibitor, UD98059 (10 µM in recording pipette solution). The value of peak amplitude of AMPAR-mediated eEPSC was normalized as percent by the value at −60 mV. The dotted line represents a linear regression of AMPAR-EPSCs in POMC neurons between −80 and 0 mV at 0 min after application of UD98059. (**b**) I-V relationship of AMPAR-EPSCs in POMC neurons in fasted mice at 0 (blue), 10 (box), and 15 (red) min after application of UD98059. (**c**) RIs of AMPAR-EPSCs in POMC neurons in fed (blue) and fasted (red) mice treated with UD98059. (**d, e**) I-V relationship of AMPAR-EPSCs and (**f**) RI in POMC neurons excluded poor space clamp recordings did not change from original results. (mean ± SEM. * and ** indicates p<0.05 and 0.01, respectively, paired t-test.).

# Materials and method

## Animals

POMC-GFP mice (21–40 days old, on a C57BL6 genetic background) were group housed and maintained on a 12–12 hr light/dark cycle with food and water available ad libitum. The fasting was performed overnight before the experiment. Leptin and rapamycin were injected (IP) at 6:30 pm the day before electrophysiological experiments. HFD, which contained 45% kcal from fat (Rodent Chow #D12451, Research Diets), was given for 10 days after weaning. All animal procedures were performed in strict accordance with NIH Care and Use of Laboratory Animals Guidelines and were approved by the Yale University Animal Care and Use Committee.

## Electrophysiology

The preparation of hypothalamic slices containing POMC and NPY neurons was performed as described previously (*Suyama et al., 2016b*). Briefly, after mice were anesthetized with isoflurane and decapitated, the brains were rapidly removed and immersed in cold (4°C) carboxygenated high-sucrose solution containing (mM): sucrose 220, KCl 2.5, NaH2PO4 1.23, NaHCO3 26, CaCl21, MgCl2 6 and glucose 10, pH 7.3 with NaOH).

After being trimmed to a small tissue block containing the hypothalamus, coronal slices (300 μm thick) were cut on a vibratome and maintained in a holding chamber with artificial cerebrospinal fluid (ACSF, bubbled with 5% CO2% and 95% O2) containing (in mM): NaCl 124, KCl 3, CaCl$_2$ 2, MgCl$_2$ 2, NaH$_2$PO$_4$ 1.23, NaHCO$_3$ 26, glucose 10; pH 7.4 with NaOH). After a 1 hr recovery period, slices were transferred to a recording chamber and were constantly perfused with ACSF (33°C) at a rate of 2 ml/min.

Whole-cell patch clamp recording was performed in POMC and NPY-GFP neurons under voltage clamp as reported previously (*Suyama et al., 2016a*). The micropipettes were made of borosilicate glass (World Precision Instruments) with a Sutter micropipette puller (P-97) and back-filled with a pipette solution containing (mM): Cs-Methaneslufonate 135, MgCl2 2, HEPES 10, EGTA 1.1, Mg-ATP 2.5, Na$_2$-GTP 0.3, and Na$_2$-phosphocreatin 10, pH 7.3 with KOH. Both input resistance and series resistance were monitored throughout the experiments and the former was partially compensated. Only those recordings with stable series resistance and input resistance were accepted. All data were sampled at 3–10 kHz, filtered at 1–3 kHz and analyzed with an Apple Macintosh computer using Axograph X (AxoGraph).

For mEPSC recording, 0.5 μM TTX and 50 μM picrotoxin were added to aCSF and holding potential was −60 mV. For measurement of AMPAR rectification and AMPA-NMDA ratio, 3.5 mM QX-314, 5 mM tetraehylanmonium-Cl and 0.1 mM spermine were added to the pipette solution, and picrotoxin were added to aCSF. Peak amplitude of eEPSCs were measured at −80 to +60 mV holding potentials in the presence or absence of 50 μM D-AP5 in aCSF. NMDAR-mediated eEPSC was calculated by the subtraction of the eEPSC value in the presence of D-AP5 from this value in the absence of D-AP5.

The value of peak amplitude of AMPAR-mediated eEPSC was normalized as percent by the value at −60 mV for the analysis of AMPAR-rectification. AMPAR-rectification index was calculated by a value at 40 mV divided by the value at −60 mV.

## Beta-endorphin secretion measurement

Male mice fed a regular mouse chow diet [(4.8% of fat) (Specialty Feeds, Glen Forrest, Australia)] were randomly split into two groups of either fed or fasted. Within these two treatment groups, mice were further divided into four treatments, in which the hypothalamic explants would be treated. These groups were: AMPA (10 μM) (Tocris bioscience, Cat NO. 0169), AMPA (10 μM) (Tocris bioscience, Cat NO. 0169) + IEM 1460 (100 μM) (Tocris bioscience, Cat NO. 1636), IEM 1460 (100 μM) (Tocris bioscience, Cat NO. 0169), or aCSF.

At 10 weeks of age, the fasted treatment group were deprived of food for 48 hr before the beginning of the experiment. For full methods of secretion experiments refer to (*Enriori et al., 2007*). Mice were sacrificed by decapitation and the brains were immediately removed. A block of the hypothalamus was cut and a 2 mm slice of the mediobasal forebrain was made using a vibrating microtome (Leica VS 1000, Leica micro systems, Inc., Bannockburn, IL). Each slice was incubated first

with aCSF treatment in an incubator at 37°C, 95% O2% and 5% CO2 for 1 hr. This was followed by another incubation in aCSF, followed by an incubation in the assigned treatment as outlined above, and finally an incubation in 56 mM of KCL. Each incubation period lasted 45 min. Following each 45 min incubation, the supernatant was removed and frozen for future RIA analysis.

The $\beta$ endorphin RIA was performed following instructions of the Phoenix Pharmaceutical RIA kit (Phoenix Pharmaceuticals, Beta endorphin (rat) Cat# RK-022–33).

Results are expressed mean $\pm$ SEM. Results were analyzed by two-way analysis of variance (ANOVA) and Tukey multiple comparison posthoc test.

## Acknowledgement

The authors thank Marya Shanabrough, Klara Szigeti-Buck and Jeremy Bober for technical support. This work was supported by the US National Institutes of Health (DK111178, AG052986, AG051459, DK097566, DK105571 and DK107293), the American Diabetes Association, The Klarman Family Foundation and the Helmholtz Society (ICEMED). SS was supported by a fellowship from Manpei Suzuki Diabetes Foundation.

## Additional information

### Funding

| Funder | Grant reference number | Author |
|---|---|---|
| National Institutes of Health | DK045735 | Marcelo O Dietrich |
| National Institutes of Health | DK107916 | Marcelo O Dietrich |
| National Health and Medical Research Council | | Stephanie E Simonds |
| Monash University | | Stephanie E Simonds |
| National Heart Foundation of Australia | | Stephanie E Simonds |
| National Health and Medical Research Council | 1065641 | Michael A Cowley |
| National Health and Medical Research Council | 1079422 | Michael A Cowley |
| National Institutes of Health | R01 DK097566 | Sabrina Diano |
| National Institutes of Health | R01 DK105571 | Sabrina Diano |
| National Institutes of Health | R01 DK107293 | Sabrina Diano |
| National Institutes of Health | AG051459 | Tamas L Horvath |
| National Institutes of Health | AG052986 | Tamas L Horvath |
| National Institutes of Health | DK111178 | Tamas L Horvath |

The funders had no role in study design, data collection and interpretation, or the decision to submit the work for publication.

### Author contributions

SS, Conceptualization, Formal analysis, Funding acquisition, Investigation, Writing—original draft, Project administration, Writing—review and editing; AR, Conceptualization, Data curation, Formal analysis, Investigation, Methodology, Writing—review and editing; Z-WL, Writing—review and editing; MOD, Data curation, Formal analysis, Methodology; TY, Conceptualization; SES, Conceptualization, Data curation; MAC, Data curation, Investigation; X-BG, Conceptualization, Writing—original draft; SD, Data curation, Formal analysis, Writing—original draft; TLH, Conceptualization, Funding acquisition, Writing—original draft

### Author ORCIDs

Tamas L Horvath, http://orcid.org/0000-0002-7522-4602

**Ethics**

Animal experimentation: All animal procedures were performed in strict accordance with NIH Care and Use of Laboratory Animals Guidelines and were approved by the Yale University Animal Care and Use Committee (protocol number 2017-07942) and Monash University (SOBSA/P2009/37).

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
