## [Decision Letter]

Thank you for submitting your article "Plasticity of calcium-permeable AMPA receptors in POMC neurons" for consideration by *eLife*. Your article has been reviewed by three peer reviewers, one of whom is a member of our Board of Reviewing Editors and the evaluation has been overseen by Richard Aldrich as the Senior Editor. The reviewers have opted to remain anonymous.

The reviewers have discussed the reviews with one another and the Reviewing Editor has drafted this decision to help you prepare a revised submission.

Summary:

Suyama and colleagues describe results from a series of studies investigating the role and regulation of AMPA receptors in POMC neurons during fed and fasted conditions. The overall hypothesis is that in the fed state (or after leptin treatment), the AMPA receptor is composed primarily of GluR1 is rectifying and conducts calcium. In contrast, while in the fasted state the channel is composed of GluR1 and GluR2 conducts less calcium and is not rectifying. Briefly, they use an array of approaches to assess the effects of inhibition of AMPA receptors on POMC neuronal properties and EPSCs. They found that AMPA antagonism blunted EPSCs in the fed but not the fasted state. In addition, exposure to high fat diet and elevated leptin levels increase POMC neuronal activity and increased expression of AMPA receptors. It is notable that all of the rectification indices are considerably less than 1, indicating a substantial contribution for calcium-permeable AMPA receptors in POMC neurons. Overall, the studies are well described and provide data that will be of wide interest. The findings are consistent with prior literature showing plasticity of neurons under different nutritional conditions but add important new information by showing correlated with changes in the AMPA receptor composition.

Essential revisions:

The recordings leave much to be desired as convincing evidence of changes in synaptic current rectification. Measurement of rectification index (RI) requires good voltage clamp, especially at depolarized potentials. AMPA-R currents should show a reversal potential close to 0 mV. However, based on the IV relationships reported here, many of the cells show reversal potentials that are substantially different from 0 mV. This is especially concerning when comparing the ratio of synaptic current amplitudes at +40mV and -60 mV where different reversal potentials are evident in the IV curves. This problem also manifests as variability in the RI of between ostensibly similar conditions various (Fed condition: 0.3, 0.45, 0.4, 0.426). This is likely due to changes in other conductances (e.g., K-conductances) associated with the various experimental manipulations that make cells more or less difficult to voltage clamp. Use of CsCl and QX-314 in the internal solution is frequent, to aid in achieving effective voltage clamp at depolarized potentials.

Some more detail regarding the specificity of the AMPR receptor antisera is required. It is unclear if the authors are citing the use of the antisera or just the methods being used for immunohistochemistry. The complete absence of GluR2 immunostaining in Figure 6 is concerning. Removal of the figure would help the manuscript.

It is implicit in the authors' hypothesis that calcium conductance by the channel with increased levels of GluR1 (and less GluR2) should result in increased POMC activation as compared to the GluR1 + GluR2 channel in the fasted state. Is there any evidence to support this? Are there precedents in the literature showing that this form of the AMPA receptor can confer an increase of the activity of other neuronal types?

In the Discussion the authors state that the synaptic efficiency of the receptor is higher. It is not entirely clear what this means and what evidence there is to support this assertion.

Consistent with their hypothesis, the authors show effects of greater effects of IEM-1640 on AMPA receptors in the fed rather than the fasted state and that CNQX and IEM-1640 block EPSCs altogether. What is the effect of CNQX alone? The prediction is that if anything it should have the opposite effect.

---

## [Author Response]

*Essential revisions:*

*The recordings leave much to be desired as convincing evidence of changes in synaptic current rectification. Measurement of rectification index (RI) requires good voltage clamp, especially at depolarized potentials. AMPA-R currents should show a reversal potential close to 0 mV. However, based on the IV relationships reported here, many of the cells show reversal potentials that are substantially different from 0 mV. This is especially concerning when comparing the ratio of synaptic current amplitudes at +40mV and -60 mV where different reversal potentials are evident in the IV curves. This problem also manifests as variability in the RI of between ostensibly similar conditions various (Fed condition: 0.3, 0.45, 0.4, 0.426). This is likely due to changes in other conductances (e.g., K-conductances) associated with the various experimental manipulations that make cells more or less difficult to voltage clamp. Use of CsCl and QX-314 in the internal solution is frequent, to aid in achieving effective voltage clamp at depolarized potentials.*

We understand the importance of correct voltage in the measurements of IV curves of synaptic currents. Errors in voltage reported in this manuscript could have originated from liquid junction potential (LJP), series resistance (Rs), opening of K^+^ channels at positive membrane potential and poor space clamp, etc. Many of these factors could be compensated for (such as LJP, Rs and opening of K^+^ channels) but the poor space clamp is, at present, a limitation in electrophysiological recordings from slice preparations. We have compensated for LJP and Rs; also, we used Cs^+^ instead of K^+^ in the pipette solution (Please note that we corrected these points in the Materials and methods section of the revised manuscript). The main reason for the error in voltage described in this manuscript is probably due to poor space clamp, which is different in individual cells recorded in this study. Therefore, we have re-examined all of the recorded cells and excluded recordings in which reversal potential deviated significantly from 0 mV likely representing poor space clamp, and, we re-calculated the rectification index (RI). After doing so, the RI values were similar to the original results and were still significantly different between control and test groups as they were in the original results. Therefore, since the space clamp problem existed equally in all measurements among all groups, our original conclusions were not affected by this issue. The re-analysis also showed the constant RI values in fed condition as 0.32-0.35, except ND fed condition (RI was 0.48) of HFD vs. ND experiment. Since HFD vs. ND experiment was performed with 4-5 weeks old mice while the other experiments was with 3 weeks old mice, the RI in fed condition could be different depend on the age. We have provided these results in the text (subsections “Leptin suppresses GluR2 in the AMPA receptor complex of POMC neurons” and “ERK promotes GluR2 in the AMPA receptor complex of POMC neurons” and Discussion section).

*Some more detail regarding the specificity of the AMPR receptor antisera is required. It is unclear if the authors are citing the use of the antisera or just the methods being used for immunohistochemistry. The complete absence of GluR2 immunostaining in Figure 6 is concerning. Removal of the figure would help the manuscript.*

In accordance with the reviewer’s comment, we have removed immunohistochemistry data from the manuscript.

*It is implicit in the authors' hypothesis that calcium conductance by the channel with increased levels of GluR1 (and less GluR2) should result in increased POMC activation as compared to the GluR1 + GluR2 channel in the fasted state. Is there any evidence to support this? Are there precedents in the literature showing that this form of the AMPA receptor can confer an increase of the activity of other neuronal types?*

It is well known that increasing GluR2-lacking (=Ca^2+^ permeable) AMPAR promotes synaptic plasticity such as LTP and LTD dependent on neuron subtypes.We acknowledge that direct evidence showing increased activation of POMC neurons by increased levels of GluR2-lacking receptors is still missing, particularly in whole animals. However, in our case, in POMC neurons, mEPSC amplitude in the fed state was larger than in the fasted state, whereas in the absence of extracellular Ca^2+^, mESPC amplitude was comparable in the fed and fasted states. These results imply that POMC neurons would have an enhanced excitatory input and Ca^2+^ influx under the fed state, which would be translated into enhanced output of the POMC system.

*In the Discussion the authors state that the synaptic efficiency of the receptor is higher. It is not entirely clear what this means and what evidence there is to support this assertion.*

We acknowledge that this statement was not clear. We have revised this statement in the Discussion as follows:

“we showed that mEPSC amplitude which reflects efficiency of the synaptic transmission at the postsynaptic site is higher in POMC neurons in the ARC in fed animals than in fasted animals, […]. Alternatively, it can also be interpreted as a depression of efficiency of glutamatergic transmission onto POMC neurons during food restriction.**”**

*Consistent with their hypothesis, the authors show effects of greater effects of IEM-1640 on AMPA receptors in the fed rather than the fasted state and that CNQX and IEM-1640 block EPSCs altogether. What is the effect of CNQX alone? The prediction is that if anything it should have the opposite effect.*

CNQX alone should block all AMPAR-mediated EPSCs. Application of CNQX after IEM-1640 excluded the possibility of contamination by currents other than AMPAR-mediated EPSCs.